Cascading effects of a highly specialized beech-aphid–fungus interaction on forest regeneration

Cook-Patton Susan C. 1 cook-pattons@si.edu
Maynard Lauren 1
Lemoine Nathan P. 2
Shue Jessica 1
Parker John D. 1
1 Smithsonian Environmental Research Center , Edgewater, MD , United States
2 Florida International University , United States
Huang Xiaolei
Electronic publication date: 2014 Jun 17
Publication date: 2014
Volume: 2
Electronic Location ID: e442
Received 2014 Apr 1; Accepted 2014 Jun 2
Copyright: © 2014 Cook-Patton et al.
Copyright year: 2014
Copyright holder: Cook-Patton et al.
License: This is an open access article, free of all copyright, made available under the Creative Commons Public Domain Dedication. This work may be freely reproduced, distributed, transmitted, modified, built upon, or otherwise used by anyone for any lawful purpose.
License URL: https://creativecommons.org/publicdomain/zero/1.0/

Keywords: Seedling survival, Grylloprociphilus imbricator, Scorias spongiosa, Forest regeneration, Fagus grandifolia, Specialist herbivore, Indirect interactions

Funding: Washington Biologists Field Club NSF-REU grant DBI 156799 A grant from the Washington Biologists Field Club to SCC and JDP, and an NSF-REU grant to JDP (DBI 156799) supported this research. The funders had no role in study design, data collection and analysis, decision to publish, or preparation of the manuscript.

==============================
Specialist herbivores are thought to often enhance or maintain plant diversity within ecosystems, because they prevent their host species from becoming competitively dominant. In contrast, specialist herbivores are not generally expected to have negative impacts on non-hosts. However, we describe a cascade of indirect interactions whereby a specialist sooty mold (Scorias spongiosa) colonizes the honeydew from a specialist beech aphid (Grylloprociphilus imbricator), ultimately decreasing the survival of seedlings beneath American beech trees (Fagus grandifolia). A common garden experiment indicated that this mortality resulted from moldy honeydew impairing leaf function rather than from chemical or microbial changes to the soil. In addition, aphids consistently and repeatedly colonized the same large beech trees, suggesting that seedling-depauperate islands may form beneath these trees. Thus this highly specialized three-way beech-aphid–fungus interaction has the potential to negatively impact local forest regeneration via a cascade of indirect effects.

Introduction

Natural enemies play important roles in structuring plant communities (Janzen, 1970; Webb & Peart, 1999; Carson & Root, 2000; Petermann et al., 2008). While specialist enemies (i.e., those that feed on one or a few closely related species) can in some cases remove entire species from the plant community (e.g., Herms & McCullough, 2014), they are generally thought to have positive effects on plant biodiversity because they prevent any one species from becoming competitively dominant (Janzen, 1970; Connell, 1971; Terborgh, 2012). Similarly, when specialist enemies are lost, which can occur when a plant occupies a new range, that plant species may become invasive and outcompete native biodiversity (Mack et al., 2000; Keane & Crawley, 2002). In contrast, it is not generally expected that specialist enemies will negatively impact the growth and survival of non-host plant species. However, ecological systems often consist of complex interaction webs, and species may impact ecologically distant species indirectly via changes in either the densities or the traits of intermediary species (Abrams, 1995). Thus, it is theoretically possible for specialist enemies to have cascading negative effects on the broader biotic community.

Here, we document the negative effects of a highly specialized three-way plant-herbivore-fungal interaction on non-host plant species. Specifically, we examined the effect of wooly beech aphid (Grylloprociphilus imbricator (Fitch)) colonization on forest seedling communities. Wooly beech aphids (Fig. 1A) are common consumers of American beech trees (Fagus grandifolia) in eastern North American forests (Hottes & Frison, 1931; Smith, 1974; Blackman & Eastop, 1994; Aoki, Kurosu & von Dohlen, 2001). Known colloquially as “boogie-woogie aphids” for their tendency to shake their abdomen when disturbed, these aphids are frequently discussed in environmental blogs and state extension publications (e.g., Childs, 2011; Virginia Department of Forestry, 2013). However, their natural history is only barely described, and their ecology and impacts on co-occurring species is to our knowledge entirely uncharacterized.

Figure 1 Wooly beech aphids and its fungal specialist.

(A) Wooly beech aphids, (B) aphid colony covering a beech branch, (C) Scoria spongiosa before it turns black, and (D) blackened S. spongiosa on the leaves of a seedling beneath an infested beech tree.

These wooly, white aphids form highly conspicuous colonies (Fig. 1B) on branches of beech trees. The colonies can be up to 1.5 m in length and contain thousands of individuals (Hottes & Frison, 1931; Smith, 1974; Blackman & Eastop, 1994; Aoki, Kurosu & von Dohlen, 2001). A single, wingless mother (or “fundatrix”) starts a colony after hatching from an over-wintering egg (Smith & Denmark, 1984) and in North Carolina, this fundatrix and her parthenogenic offspring can be found from April until November on beech trees (Smith & Denmark, 1984). A second generation of winged females (“sexuparae”) may appear anytime between June and the end of November (Smith & Denmark, 1984; Aoki, Kurosu & von Dohlen, 2001). In Maryland where we conducted our research the aphid colonies may appear as early as May, but do not become common until late August/early September (S Cook-Patton, pers. obs., 2014).

The aphid colonies are made more obvious by the fungal masses that form below them (Fig. 1C). This fungus (Scorias spongiosa (Schwein.) Fr.) specializes on the aphid’s sugar-rich excrement or “honeydew” (Hughes, 1976). S. spongiosa is found primarily in association with Fagus species (Reynolds, 1978), but also on Alnus species (Chomnunti et al., 2011). Initially the fungus forms a brown, spongy mass that eventually turns black, hardens, and persists through much of the winter. The fungus also coats the leaves of seedlings directly beneath the aphid colonies (Fig. 1D).

We explored the factors determining aphid distributions across the forest landscape at multiple spatial scales, as well as the consequences of aphids for the forest seedling community, by combining two years of field observations in a mapped 16-ha forest with a common garden experiment. We tested three principal hypotheses: (1) the distribution of aphids across the landscape will be non-random, (2) the beech-aphid-sooty mold interaction will have negative effects on seedling communities and (3) this negative effect will result from changes in soil quality. Based on our initial observations, we predicted that aphids would be most common on small trees in sunny patches and theories of negative density dependence suggested that specialist herbivores would also be more common in locations with high beech density (Janzen, 1970; Connell, 1971). We predicted that seedlings beneath heavily infested branches would have increased mortality, based on our initial observations of seedling die-off in the forest, and that aphid honeydew was reducing seedling survival because the carbon-rich inputs diminished soil fertility (Stadler, Michalzik & Müller, 1998; Blumenthal, Jordan & Russell, 2003).

Methods

Experimental site

All field surveys and soil collections occurred in the Smithsonian Environmental Research Center (SERC) Forest Dynamics Plot (38°53′11.4822″, −76°33′31.2464″). In this 16-ha plot, which is further divided into 100 10 m × 10 m subplots, the diameter and spatial location of all woody species >1.0 cm dbh are known. We censused every beech (N = 659 trees) occurring within 12–13 evenly spaced subplots per hectare (Fig. 2A; N = 204). For a random subset of the subplots (N = 258, N = 129 per year), we also collected light availability and soil moisture data. We gathered light data in August 2011 using an AccuPAR LP-80 ceptometer to record photosynthetically available light in the center of each plot, taking all measurements between 11am and 4pm on a mostly cloudless day. We then collected ambient light measurements from a nearby, unshaded area and calculated ‘light transmittance’ as the fraction of light in each forested location relative to ambient light. We collected soil moisture data in June 2011 using a Fieldscout TDR 300, with two soil moisture measurements taken from the southwest corner of each 10 m × 10 m subplot.

Figure 2 Spatial distribution of beech trees and aphids.

(A) Plot map with the number of beech trees in each subplot indicated by circle size and the number of aphid-infested trees indicated by color. Blank areas within the regular grid represent plots without beech trees, including the curved area from top right to bottom left where a stream occurs. (B) Spatial clustering at different spatial scales. The x-axis represents the mean distance within a distance class. Points outside of the dashed confidence interval are significantly different than random, with points above showing significant clustering.

Field survey

During the last week of September 2012 and the third week of October 2013, we recorded aphid infestation by visually scanning beech trees from the base up to ∼20 m. Most colonies occurred within the first 10 m (S Cook-Patton, pers. obs., 2012 & 2013), but it is possible that additional, unseen colonies occurred much higher in the canopy. Aphids packed densely along a single branch or a cluster of branches. We therefore scored infestations with an ordinal scale: 0 = no aphids, 1 = presence of aphids, 2 = greater than 30 cm of branch covered, and 3 = greater than 100 cm covered. In the second week of October 2012, we also selected 19 focal beech trees that occurred within the northwest hectare of the SERC Forest Dynamics Plot. We chose trees that were 10–20 cm dbh, because the trees of this size class were sufficiently large (only 7% of censused trees were bigger), common, and varied in aphid infestation status. Six of the focal beech trees had no aphids, seven had an aphid score of two, and six had an aphid score of three. We tagged and identified every woody seedling within 1 m of the focal tree (N = 575 total; 30 per adult on average, ranging from 7 to 58), measured its initial height, and noted whether its leaves were directly coated in sooty mold. In mid-June 2013, we returned to these seedlings to reassess height and survival.

Common garden experiment

We collected the top ∼20 cm of soil in mid-May 2013 from randomly selected, but similarly sized adult beech trees that occurred in the northwest corner of the SERC Forest Dynamics Plot (N = 15 adults with aphids in 2012, 15 adults without aphids). For trees with aphids, we gathered soil from directly beneath the previous year’s infestation (within 0.5 m of the tree trunk). These areas were obvious because they had fewer seedlings, blackened leaf litter, and black-encrusted branches directly above. For trees without aphids, we gathered soil from an analogous position within 0.5 m of the tree trunk. We divided the soil from each adult tree into six, sterilized tree tubes (Deepots, Recycled D40 cells; sterilized with 10% bleach, 10 min) and sterilized all tools between samples. Into each tube we planted a one-year old seedling that naturally germinated in an adjacent, aphid-free forest patch. We employed six different species (Acer rubrum, Carya alba, Fagus grandifolia, Liriodendron tulipifera, Platanus occidentalis, and Quercus alba) and ensured that all species occurred once in each soil sample. Seedlings grew suspended outdoors in elevated trays (Deepot N25T) and beneath two layers of shade cloth, and were watered between May and August 2013. We measured initial seedling height in May 2013, and in the third week of October 2013, we assessed height and survival.

Statistical Analyses

Spatial distribution

All analyses were conducted in R (v 3.0.2). To determine the local factors that influenced aphid infestation, we examined how the probability of aphid infestation depended on tree size (dbh), year (2012 vs. 2013), the number of beech trees within a subplot, light availability, and soil water content. For all analyses we used binomial logistic regression (glm in R package stats, R Core Team). The first model included probability of infestation as a function of tree dbh and year, plus their interaction. The second model regressed probability of infestation against the number of beech trees in each subplot. In addition, we examined whether the severity of infestation in 2012 (aphid score of 0, 1, 2, or 3) determined the probability of infestation in 2013. Finally, the third model included probability of infestation as a function of percent light transmittance and soil water content for subplots where environmental data were collected.

To examine the spatial scale at which aphid outbreaks occurred in both 2012 and 2013, we calculated Moran’s I for different distance classes. We first determined the proportion of infested trees in each subplot, and then utilized Euclidean distances between subplot centroids (Fortin & Dale, 2005) to place subplots into 15 unique classes, ranging from 0 to 500 m. For example, the first distance class consisted of all grids between 0 and 36 m apart and the second distance class consisted of all grids between 37 and 71 m apart. We then calculated Moran’s I for each distance class, using 999 random permutations of the data to determine the significance of the observed test statistic (Fortin & Dale, 2005), and considered statistics falling outside of the 95% confidence interval as significant.

Forest seedling performance

We used generalized linear mixed effects models (glmer in R package lme4 (Bates, Maechler & Bolker, 2013)) to determine whether seedling growth and survival varied as a function of tree infestation status (aphid vs. no-aphid), treating survival as a binary response variable with a binomial distribution and growth as a continuous response variable with a normal distribution. We constructed a second model to determine how survival and growth varied as a function of honeydew coverage, where honeydew coverage was a three-factor predictor (seedlings beneath uninfested trees, seedlings beneath infested trees but not directly covered in sooty mold, and seedlings directly covered in sooty mold). For both models, we treated adult tree as a random factor with seedling nested within adult.

Common garden experiment

Because the seedlings species could be grouped into early successional species (Acer rubrum, Liriodendron tulipifera, and Platanus occidentalis) and late successional species (Carya alba, Fagus grandifolia, and Quercus alba), we planned a priori contrasts to determine whether the effects of soil treatment (aphid vs. no-aphids) differed depending on successional status. We also used planned a priori contrasts to assess whether the growth of beech seedlings differed from other species. We again analyzed seedling survival and growth in the common garden experiment using generalized linear models, with soil treatment, species successional category, and conspecific status (beech vs. not beech) as the predictors.

Results

Spatial distribution

Even though aphid infestations were significantly lower in 2013 than in 2012 (n = 1288, z = 8.26, p < 0.001), patterns of colonization were consistent between years. Aphids were more likely to infest trees that had hosted aphids in the previous year (n = 644, z = 8.26, p < 0.001). Trees that had few to no aphids in 2012 had a very low probability of hosting aphids in 2013. In contrast, trees with moderately sized aphid densities in 2012 (aphid score = 2) had a significantly higher probability of being infested in 2013 (0.301 ± 0.050, probability ± SE; z = 7.51, p < 0.001), and trees with severe infestations in 2012 had a very high probability of being recolonized in 2013 (0.889 ± 0.074, probability ± SE; z = 7.07, p < 0.001).

Aphids occurred on all sizes of beech trees, ranging from saplings to large adults, and we observed large infestations (aphid score = 3) on trees with dbh values ranging from 3.3 to 55.6 cm in 2012, and 3.8 to 32.3 cm in 2013. However, aphids were more common on larger beech trees, with the probability of infestation increasing with dbh (n = 1288, p < 0.0001). Aphids did not appear to recruit to denser patches of beech trees, as the proportion of beech trees infected in each subplot did not depend on the number of beech trees in that subplot (n = 262, t = 7.20, p = 0.223). Infestation did depend on the number of infested neighbors, however, as can be seen by significantly clustered Moran’s I values at small spatial scales (Fig. 2B).

Aphid infestation was unrelated to either light (n = 258, p = 0.678) or soil moisture content (n = 258, p = 0.890). Aphids may be responding to other environmental variables, however, because aphid infestations were spatially clumped at small distances (0–36 m). This clustering disappeared at larger spatial scales, however, with the distribution of infestation becoming random after our first distance class (Fig. 2B).

Forest seedling performance

Aphid colonization had negative consequences for the seedling community. When aphids were present, we always observed the co-occurrence of S. spongiosa suggesting that the aphids either carry the fungus with them, or that the fungus is ubiquitous (but dormant) in the environment. In the absence of aphids and fungus, seedling survival was very high: 90% ± 2.7% of the tagged seedlings survived between 2012 and 2013 (mean ± SE). In contrast, seedlings directly covered in honeydew/sooty mold had significantly lower survival (80 ± 4.2%, mean ± SE; n = 575, X2 = 7.40, p = 0.030; Fig. 3). This effect appears to be fairly localized, as seedlings that occurred under infested trees but outside of the honeydrew drip zone did not show diminished survival (89 ± 2.4%, mean ± SE; z = 0.51, p = 0.611). Thus, seedling survival did not depend on whether aphids occurred on the nearest adult tree, but instead on whether they were covered with honeydew/sooty mold (z = 2.18, p = 0.030; Fig. 3). Growth of the surviving seedlings showed the same patterns as survival, but did not significantly differ among aphid (n = 470, F = 0.48, p = 0.493) or honeydew conditions (n = 470, F = 0.57, p = 0.571). Beech seedlings also did not have different growth or survival than other seedling species (p > 0.1 for all tests), suggesting that they experienced little to no negative feedback from adult beech trees in the field.

Figure 3 Forest seedling performance.

Probability of seedling survival beneath an uninfested tree, beneath an aphid-infested tree but not in the honeydew drip zone, or directly beneath a honeydew drip zone.

Common garden experiment

The effect of aphids on seedling survival and growth does not appear to be mediated through changing soil conditions, because seedling survival was the same in soil from beneath infested and uninfested beech trees (n = 178, survival: z = 0.22, p = 0.603; growth: t = −0.38, p = 0.665). Otherwise, patterns of growth were consistent with other ecological predictions that early successional species would have higher growth rates than late successional species (t = −5.99, p = 0.001), and that conspecific beech seedlings would have reduced growth compared to other species (with significantly lower growth than Acer rubrum (t = 6.73, p < 0.001) and P. occidentalis (t = 4.37, p < 0.001)).

Discussion

Although wooly beech aphids are a common herbivore on American Beech trees in eastern North America, their ecology is almost entirely unknown. Here, we show that a three-way interaction among a specialist aphid, its host tree, and a fungal specialist on honeydew negatively impacts forest regeneration. We observed that tree seedlings, regardless of species identity, suffered elevated mortality when they were positioned directly under aphid colonies. However, in a common garden experiment seedlings grown in soil from infested and uninfested trees performed similarly, indicating that seedling mortality beneath aphid colonies resulted not from changes in soil quality, but from the sooty mold impairing leaf function. The aphids also preferentially and repeatedly colonized the same large beech trees, suggesting that negative effects of this highly specialized aphid-beech-fungal interaction are creating islands of seedling-depauperate patches.

There are two avenues by which the beech-aphid-fungal interaction could impact seedling mortality: changes in soil quality and/or impairment of leaf function. Aphid honeydew represents a carbon rich input into the soil, which would alter C:N ratios and diminish soil fertility (Stadler, Michalzik & Müller, 1998; Blumenthal, Jordan & Russell, 2003). We thus initially hypothesized that reductions in seedling performance would be due to changes in soil quality. However, when we grew seedlings from six different tree species in soil collected from directly beneath aphid colonies and from equivalent areas beneath uninfested trees, we observed no differences in seedling growth or survival. Thus, if honeydew was altering soil C:N ratios, the differences in soil nutrients were insufficient to affect seedling performance. Additionally, we only observed reduced seedling survival in the field when seedlings were directly covered in sooty mold, and not in nearby seedlings that fell outside of the honeydew drip line (Fig. 3). This highly localized effect implicates the honeydew and fungus as agents of mortality. Because the fungus is not known to infect leaf tissue (Hughes, 1976), we believe that mortality resulted from diminished photosynthesis or impaired gas exchange rather than direct fungal infection.

At large spatial scales, we observed that the aphid colonies were distributed widely and randomly throughout a 16 ha forest plot (Fig. 2), counter to our hypothesis that aphids would be distributed non-randomly. However, at small spatial scales (less than 36 m), aphid colonies were significantly clustered. Trees that were infested the previous year and/or close to other infested beech trees were more likely to be colonized. Very little is known about the dispersal and colonization patterns of wooly beech aphids. The second generation of wooly beech aphids, consisting entirely of winged females, are believed to migrate in late fall/early winter to their secondary host, the bald cypress, Taxodium distichum (L.) (Smith & Denmark, 1984; Aoki, Kurosu & von Dohlen, 2001). However, bald cypress is almost non-existent in the study area. The natural northern limit of bald cypress occurs at a site ∼50 km south of the study area at Battle Creek Cypress Swamp (Nature Conservancy, 2014), although there is also a small patch of planted bald cypress on SERC property. Aphid species are known to disperse over long distances, especially if they move up into the air stream (Compton, 2002), but the repeated colonization of trees and clustering at small spatial scales suggest that the aphids are overwintering locally, either on their primary host trees or on an unknown secondary host.

The reappearance of aphid colonies on the same tree in subsequent years may be due to limitations in wooly beech aphid dispersal or may be shaped by aphid choice for specific trees. While our preliminary observations suggested that aphids were recruiting to sunny patches, the systematic survey showed no effect of light availability on aphid colonization. Elevated light conditions might increase host photosynthesis and potentially host quality, but it would not directly increase phloem nitrogen content, and nitrogen is believed to limit most herbivorous insects (Mattson, 1980). Instead, we observed that the probability of aphid infestation increased with beech size. A larger tree might recruit an aphid colony, because it is easier to locate in the forest matrix or because it represents a better food source. We speculate that the former mechanism is more likely. Wooly beech aphid colonies stem from a single fundatrix, so colony size can serve as a proxy for host quality and we observed large aphid colonies (aphid score = 3) on even very small saplings. Thus, small trees must provide sufficient nutrients to support a large colony.

More generally, this paper provides an intriguing example of how a common, but poorly characterized suite of specialists can have strong negative effects on non-host plant species via a cascade of indirect effects. Because the negative effects of the fungus were indiscriminate with regards to species and had an overall dampening effect on seedling diversity, this example runs counter to the general ecological principal that specialist herbivores positively affect the broader plant community by preferentially suppressing their host species (Connell, 1971; Webb & Peart, 1999; Keane & Crawley, 2002; Terborgh, 2012).

We thank S McMahon, M LaForgia, K Edson and J Miguel for assistance in the field, G Parker and two anonymous reviewers for insightful critique, and G Parker for permission to use spatial and dbh data from the SERC Forest Dynamics Plot.

Additional Information and Declarations

Competing Interests

Author Contributions

The authors declare there are no competing interests.

Susan C. Cook-Patton conceived and designed the experiments, performed the experiments, analyzed the data, wrote the paper, prepared figures and/or tables, reviewed drafts of the paper.

Lauren Maynard and John D. Parker conceived and designed the experiments, performed the experiments, reviewed drafts of the paper.

Nathan P. Lemoine conceived and designed the experiments, performed the experiments, analyzed the data, prepared figures and/or tables, reviewed drafts of the paper.

Jessica Shue performed the experiments, reviewed drafts of the paper.

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
