# Peer review of "Cascading effects of a highly specialized beech-aphid–fungus interaction on forest regeneration"

_PeerJ, doi:10.7717/peerj.442_

## Round 0.1 · original submission · Minor Revisions

The reviewers think the paper is interesting and important, while they also suggest the paper should be improved by addressing the points they raised (see comments from the reviewers, especially reviewer2). Both reviewers suggest revisions concerning detailed information (e.g. life cycle, population ecology) on the study species and other issues in experimental design and discussion. I suggest the authors fully address the questions raised by the reviewers.

Reviewer 1 ·

Basic reporting

No Comments

Experimental design

No Comments

Validity of the findings

No Comments

Additional comments

The MS provides interesting data on ecology of aphid species Grylloprociphilus
imbricator including beech-aphid-fungus interaction.
The MS of this paper can be recommended for publication after thorough revision.

In particular:
1. Methods - Study organism lines 50 - 67 - this section is a mix of data which should be removed to introduction and partially to results.
Moreover, there is no data of biology of G. imricator - i.e. if this species is holocyclic or anholocyclic, why numerous individuals are particularly in autumn on their host plant etc. The authors have mentioned that the natural history of this species is somewhat described, however, after two years observations can add some relevant information to the biology of this species.
2. Results - the Methods part is divided in numerous sub-sections whereas in Results only Garden experiments is singled out.
3. Discussion - in comparison with length of Methods this chapter is very frugal and rather looks like comments or remarks to expanded methodology. There are also some repetitions from Introduction.
4. Figures should be cited in Results or Discussion rather than in Introduction or Methods.
5. Bates, Maechler, and Bolker 2011 line 137 is no cited in references.

Reviewer 2 ·

Basic reporting

The role of indirect effects on ecosystem processes mediated by herbivorous insects and subsequent trophic inter-linkages is still little explored. Consequently, the manuscript of Cook-Patton et al. reports the results of an important experiment which could make a significant contribution to the literature and should be of broad interest to the readership of PeerJ.
The ms provides a useful insight into the three-way plant-herbivore-fungus interaction negatively affecting non-host tree species. In detail, the experiment tested the effects of honeydew excretions from a specialist beech aphid and a specialist honeydew-colonizing fungus on seedling communities underneath infested beech trees. Two years of field observations at a forest site and a common garden experiment were used to test the underlying mechanisms. The ms is well organized and well written.
However, some points appear too general and need clarification. Moreover, I miss some more detailed information on the study organisms and some parameters, which were recorded, but were not regarded in the hypotheses. These points need to be addressed prior to acceptance for publication in PeerJ.

Abstract
- Line 13: “Specialist herbivores are often thought to benefit the larger plant community, because…” What is meant by the term “benefit”? A higher plant biodiversity? More efficient nutrient cycling? A higher resilience towards ecosystem disturbances? Please specify (see also my comments for the introduction)


Introduction
- Line 29: “…they are generally thought to be beneficial overall because they prevent any one species from becoming competitively dominant…“ Please specify the term “to be beneficial overall” which appears to be too general.


- Line 31: “…to spread aggressively and disrupt natural ecosystems..” What is meant by the term “disrupt”? The composition of the plant community, the cycling of nutrients, trophic inter-linkages, plant functional traits? Please specify.

References
Please check for alphabetically order:
Line 251: Virginia Department of Forestry. 2013.

Experimental design

In general, the experimental design is adequate to test the hypotheses. However, I miss some details on the population ecology and life cycle of the two study organisms. How do the aphids disperse? Is there a specific dispersal pattern? (Might this explain why they re-colonise the same beech trees?). By which nutrient elements are they limited (nitrogen in case of the aphid? This might help to explain why they prefer mature beech trees with differing leaf N contents or other nutrient demand compared to younger trees?). Or carbohydrates in case of the fungus? Why should they be related to light or soil moisture?
How heavy can a big aphid colony get? (Therefore, physically excluding younger trees from infestation?) Is there a phenological synchronicity between the insect, the host plant and the fungus? Is that linked to light demand or soil moisture?
- Line 74: “…, we also had light availability and soil moisture data.” The measurements of these climatic parameters (plus “light transmittance”) come a little bit “out of the blue”. Why did you decide to measure them? Its relevance is not clear and should be stated in the introduction and linked to the hypotheses as well.
These points need to be addressed prior to acceptance for publication in PeerJ.

- Line 85: “…, we recorded aphid infestation on the focal beech trees.” Just for practical reasons: How did you access the trees? Are the aphid colonies located basically on the bottom part of the trees?

- Line 102-104: “…, we gathered soil from directly beneath the previous year’s infestation (within 0.5 m of the tree trunk). These areas were obvious because they often had fewer seedlings, blackened leaf litter, and black-encrusted branches directly above.”
The soil material around tree trunks is generally highly affected by intensified element and water input fluxes via stemflow (especially for beech trees due to the crown architecture, see review by Levia and Frost 2003). I do not know whether this fact might bias the results, since not only the infested twigs above the trunk area but the whole tree canopy adds to the soil quality. Therefore, I ask the authors to comment on that.

Validity of the findings

In general, I asked the authors to include my points and questions (especially the ones concerning the study organisms in the methods part) into the discussions, which might resolve some questions behind some of the observations made (Line 213- 217: “The only factors that predicted aphid colonization were tree size, infestation status from in the prior year, and the close proximity of other infested beech trees. This suggests that the negative beech-aphid-fungal effect on seedling communities will be concentrated to localized islands across the forests, centered around large beech trees that are repeatedly colonized by the aphid.”)
The ms provides useful information on the role of indirect effects on ecosystem processes mediated by herbivorous insects and subsequent trophic inter-linkages. This is an important experiment which could make a significant contribution to the literature.

---

## Round 0.2 · accepted · Accept

All reviewers were satisified with the revised version of the manuscript and recommended publication. I therefore am glad to accept the paper for publication in PeerJ. Besides, one reviewer has attached an annotated manuscript, which may be helpful for the authors to finalize the manuscript.

Reviewer 1 ·

Basic reporting

No comments

Experimental design

No comments

Validity of the findings

No comments

Additional comments

The authors have added relevant comments to the life cycle of the studied species and the discussion has now more scientific character. I recommend the MS for publication.

Reviewer 2 ·

Basic reporting

I am very glad with the revised ms version, since the authors followed most of the reviewers´ suggestions. I therefore recommend the paper for publication in PeerJ.

Experimental design

The authors followed most of the reviewers´ suggestions and therefore improved the quality of the paper.

Validity of the findings

The authors followed most of the reviewers´ suggestions and therefore improved the quality of the paper.

Annotated reviews are not available for download in order to protect the identity of reviewers who chose to remain anonymous.